# Nickel tolerance is channeled through C-4 methyl sterol oxidase Erg25 in the sterol biosynthesis pathway

**Amber R. Matha** **, Xiaofeng Xie, Robert J. Maier, Xiaorong Lin** *

Department of Microbiology, University of Georgia, Athens, Georgia, United States of America

* xiaorong.lin@uga.edu

## Abstract

Nickel (Ni) is an abundant element on Earth and it can be toxic to all forms of life. Unlike our knowledge of other metals, little is known about the biochemical response to Ni overload. Previous studies in mammals have shown that Ni induces various physiological changes including redox stress, hypoxic responses, as well as cancer progression pathways. However, the primary cellular targets of nickel toxicity are unknown. Here, we used the environmental fungus *Cryptococcus neoformans* as a model organism to elucidate the cellular response to exogenous Ni. We discovered that Ni causes alterations in ergosterol (the fungal equivalent of mammalian cholesterol) and lipid biosynthesis, and that the Sterol Regulatory Element-Binding transcription factor Sre1 is required for Ni tolerance. Interestingly, overexpression of the C-4 methyl sterol oxidase gene *ERG25*, but not other genes in the ergosterol biosynthesis pathway tested, increases Ni tolerance in both the wild type and the *sre1Δ* mutant. Overexpression of *ERG25* with mutations in the predicted binding pocket to a metal cation cofactor sensitizes *Cryptococcus* to nickel and abolishes its ability to rescue the Ni-induced growth defect of *sre1Δ*. As overexpression of a known nickel-binding protein Ure7 or Erg3 with a metal binding pocket similar to Erg25 does not impact on nickel tolerance, Erg25 does not appear to simply act as a nickel sink. Furthermore, nickel induces more profound and specific transcriptome changes in ergosterol biosynthetic genes compared to hypoxia. We conclude that Ni targets the sterol biosynthesis pathway primarily through Erg25 in fungi. Similar to the observation in *C. neoformans*, Ni exposure reduces sterols in human A549 lung epithelial cells, indicating that nickel toxicity on sterol biosynthesis is conserved.

## Author summary

Nickel is commonly known as an allergen and toxin for humans, but the way in which nickel causes adverse effects is unknown. We sought to use *C. neoformans* as a model to investigate the primary targets of nickel and how cells tolerate this commonly occurring metal. We found that in both mammalian cells and fungal cells, exposure to nickel causes sterol deficiency. We discovered that Erg25, an essential enzyme key to the production of

**Data Availability Statement:** All relevant data are in the manuscript are and its supporting information files.

**Funding:** This work was supported by National Institutes of Allergy and Infectious Diseases (http://

www.niaid.nih.gov) (R01AI140719 to XL) and
University of Georgia Gene E. Michaels endowment
fund (to XL). The funders had no role in study
design, data collection and analysis, decision to
publish, or preparation of the manuscript.

**Competing interests:** The authors have declared
that no competing interests exist.

ergosterol (fungal equivalent of cholesterol), was critical for cryptococcal cells to tolerate
nickel. Cells unable to increase production of this enzyme in response to nickel exposure,
such as the *sre1Δ* mutant with the Sterol Regulatory Element-Binding regulator disrupted,
were incapable of growing in the presence of nickel. Therefore, it appears that both cells
react to nickel through upregulating a conserved biochemical pathway and particularly
the Erg25 enzyme. This work could guide future investigations into novel approaches to
manage nickel toxicity.

## Introduction

Ni is an abundant natural element ubiquitously found in soil and through industrial pollution
[1,2]. The concentrations of this metal vary widely in different environments and organisms
cope in various ways [2]. Unlike metals such as copper (Cu) or iron (Fe), no mammalian
enzymes require Ni as a cofactor [3]. Indeed, Ni is generally characterized as a toxic heavy
metal for humans. Exposure to Ni primarily occurs by inhalation or ingestion but also through
interaction with everyday items that contain Ni, such as jewelry, zippers, paper clips, and stain-
less steel dining flatware [4]. Additionally, corrosion of Ni-containing implants used in joint
and hip prostheses may lead to elevated Ni levels in the body [5]. Occupational exposure to
nickel is the highest for those involved in producing, processing, and using nickel [6]. A
National Occupational Exposure Survey conducted by the NIOSH agency from 1981 to 1983
estimated that 727,240 workers in the US were exposed to toxic levels of Ni (NIOSH 1990).

It is hypothesized that Ni is toxic to mammalian cells due to its ability to catalyze Fenton
chemistry, which culminates in oxidative stress to the cells [7,8], including lipid peroxidation
[9–11]. Ni is also capable of inducing calcium signaling pathways and activating HIF1-α
[12,13], which plays a central role in the progression of some cancers [14]. HIF1-α is typically
activated when cells experience hypoxia in the tumor microenvironment. This activation
causes transcriptional increases in genes associated with angiogenesis, growth factors, pH reg-
ulation, and apoptosis [15]. Thus, Ni has been characterized as a carcinogen in mammalian
systems. Additionally, Ni has been shown to induce a disturbance in testosterone synthesis
[16]. The main therapy to mitigate Ni toxicity is administration of antioxidants, such as gluta-
thione, which reduces lipid peroxidation in human lymphocytes [17]. Because of the pleiotro-
pic effects of Ni on cell structures and metabolism, it is difficult to define the primary
mechanism of nickel toxicity.

Despite the fact that Ni is abundant in the environment, little research has been done to
identify how environmental microbes tolerate this metal. *Cryptococcus neoformans*, a ubiqui-
tous environmental fungus, is a model organism for studying fungal cellular biology due to the
abundance of tools available to study and manipulate the fungus [18–21]. *C. neoformans* has
been used as a model to better understand cellular mechanisms conserved in mammals such
as uniparental mitochondrial inheritance [22,23], meiosis [24–27], epigenetic regulation
[28,29], and intercellular communication [30,31]. Here, we chose this fungus to investigate the
molecular mechanism for Ni tolerance. In *C. neoformans*, urease is the only known protein
that requires Ni for its function, although there are nine known Ni-dependent enzymes in
other microbes [32]. Despite the fact that the role of urease in cryptococcal pathogenesis is well
defined [33–36], its role in nickel tolerance is unknown.

Given that animals and fungi are closely related in the eukaryotic domain, understanding
the effect of nickel on fungi and how fungi tolerate Ni could be informative. This study aims to
identify pathways and factors critical for cryptococcal tolerance to Ni. We found that Ni

exposure reduces ergosterol levels and altered lipid profiles in *C. neoformans*. We screened the transcription factor deletion set and identified Sre1 as an essential factor for cryptococcal growth on Ni-supplemented medium. Sre1 is highly conserved across *Eukarya*, and it is known to regulate the ergosterol biosynthesis pathway (EBP) in response to hypoxia and hypoxia-mimicking conditions [37–41]. We found that nickel, in contrast to hypoxia that exerts a broad impact on cryptococcal transcriptome, more narrowly but profoundly alters expression of EBP genes. Erg25, a conserved C-4 methyl sterol oxidase, but not other Erg enzymes in the sterol biosynthetic pathway or the known nickel-binding protein Ure7, is specifically required for cryptococcal tolerance to Ni. Increasing the levels of the C-4 methyl sterol oxidase effectively mitigates Ni toxicity, suggesting that Erg25 is a primary target of Ni toxicity. We further demonstrated that exposure to Ni in mammalian cells also reduced the sterol levels, mimicking what we observed in the fungus. Thus, nickel might exert its toxicity effects by primarily targeting the conserved sterol biosynthetic pathway in both fungi and animals.

## Results

### Screening the transcription factor and kinase gene deletion libraries identified transcription factor Sre1 as required for cryptococcal tolerance of Ni

As a ubiquitous environmental fungus often found in soil, *C. neoformans* is subjected to exogenous Ni. Here we first determined the Ni tolerance level of the wildtype H99 cells and found that Ni concentrations beyond 250μM impaired growth of H99 (S1A Fig). As the only known *C. neoformans* enzyme that requires Ni is urease, we tested if urease plays a role in nickel tolerance. Nic1 imports Ni from the environment and Ure7 likely binds and delivers Ni to the apourease, Ure1 [35]. As expected, all three mutants, *ure1*Δ, *ure7*Δ, and *nic1*Δ, had abolished urease activity as indicated on Christensen Urea Agar (CUA) (S1B Fig). As urease activity was dependent on Ni, addition of Ni chelator dimethylglyoxime (DMG) [42] abolished urease activity in H99 cells (S1C and S1D Fig). However, we found neither *ure1*Δ nor *ure7*Δ were hypersensitive to exogenously added Ni (S1E Fig), indicating that urease is not involved in Ni tolerance in *C. neoformans*.

To identify factors that contribute to Ni tolerance, we screened the *C. neoformans* kinase [43] and transcription factor [44] deletion library collections (representing 155 and 129 genes, respectively) to identify mutants sensitive to Ni at 250 μM. We found the *sre1*Δ mutant was the only strain hypersensitive to Ni (Fig 1A).

Sre1, or Sterol Regulatory Element Binding Protein (SREBP) as it is known in animals, is a transcription factor well known to regulate the sterol biosynthetic pathway in response to several stimuli including hypoxia [37,45]. Sre1 exists in an inactive state as a full-length protein on the endoplasmic reticulum in coordination with the anchor protein Scp1. Upon various stimuli (hypoxia or low levels of ergosterol), this complex is trafficked to the Golgi where the protease Stp1 cleaves the N-terminus of Sre1. Once cleaved, the Sre1 N-terminus (N-Sre1) translocates to the nucleus to induce transcription of downstream target genes (Fig 1B, [38]). Indeed, all the deletion mutants of the Sre1 pathway components, namely *scp1*Δ and *stp1*Δ, were also hypersensitive to Ni (Fig 1A). Expression of the N-terminus portion of Sre1 in the *stp1*Δ background partially restored its growth on Ni (Fig 1A), consistent with the idea that activation of Sre1 is required for nickel tolerance. To confirm that Ni activates Sre1, we constructed a Sre1 allele with a FLAG tag at its N-terminus and expressed this construct in the wild type. When cultured on RPMI, only the full length Sre1 protein was detected (Fig 1C). However, when the strain was grown on RPMI media containing 250μM Ni (N) or 4μg/mL Fluconazole (F), a lower band at 75kDa consistent with the cleaved N-Sre1 became visible in

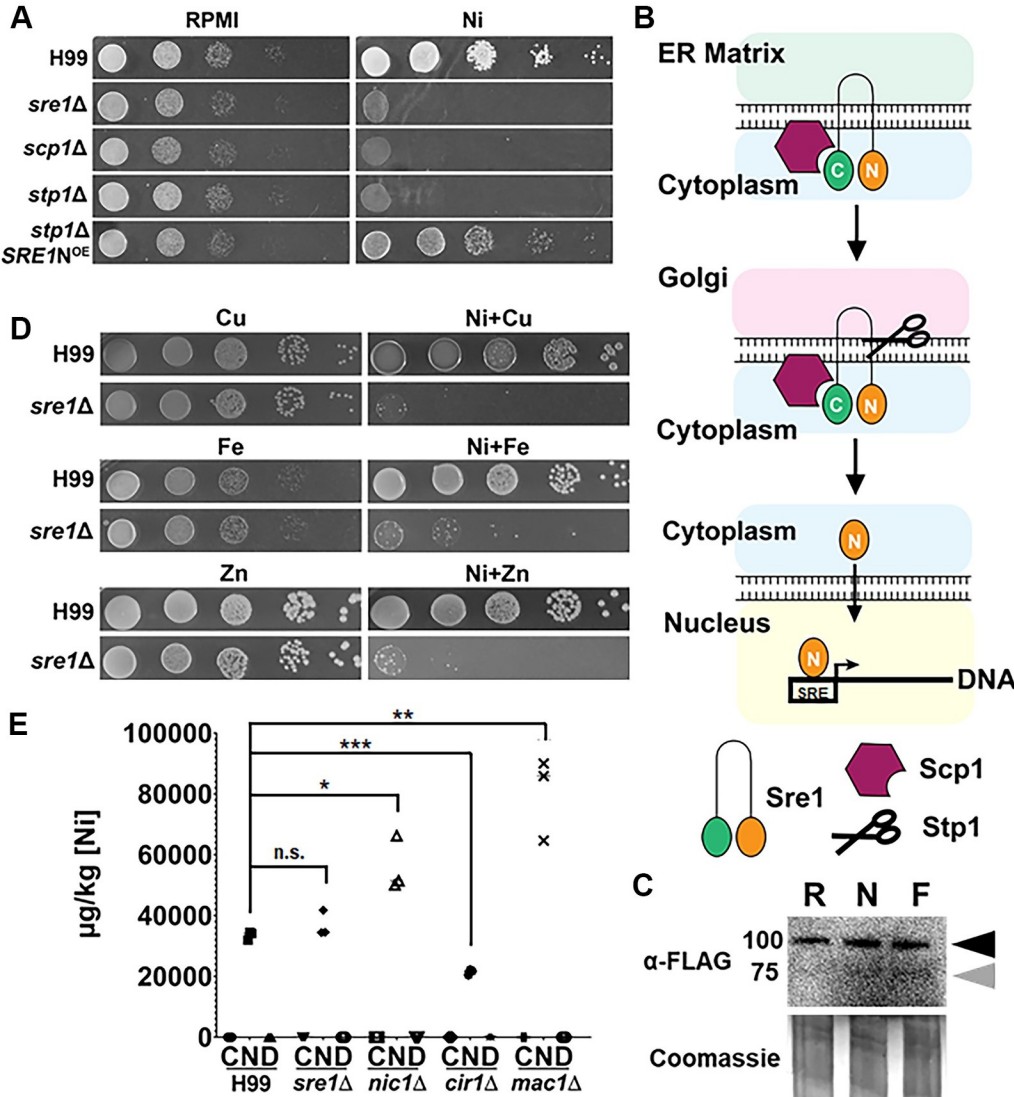

**Fig 1. *sre1Δ* growth defect is specific to Ni. (A)** Serial dilutions of H99, *sre1Δ*, *stp1Δ*, and *scp1Δ* cells were spotted onto RPMI agar ± 250 μM NiSO₄ (Ni). **(B)** Diagram of the activation process for Sre1. In response to a stimulus (e.g. drop of the ergosterol level in membrane in response to hypoxia), the full-length Sre1 residing in the ER membrane will be shuttled to the Golgi. Here, the protease Stp1 cleaves Sre1. The released N-Sre1 then translocates to the nucleus and binds the sterol regulatory element (SRE) in promoter regions of downstream targets to initiate transcription. **(C)** Cells constitutively expressing Flag-Sre1 were grown on RPMI (R), RPMI+ 250 μM Ni (N), and RPMI+ 4μg/mL Fluconazole (F). Whole-cell extracts were prepared and immunoblot analysis was done using anti-Flag purified antibody. The black and grey arrowheads denote the full-length and cleaved forms of Sre1, respectively. Coomassie staining indicates the loading. **(D)** Growth of H99 and *sre1Δ* on RPMI with the indicated metals: CuSO₄ (Cu), ZnCl₂ (Zn), and FeSO₄ (Fe) at 250 μM with or without addition of 250 μM Ni. **(E)** ICP-MS quantification of Ni concentrations in H99, *sre1Δ*, *nic1Δ*, *cir1Δ*, and *mac1Δ* cells grown on RPMI (C), RPMI+Ni (N), or RPMI+DMG (D). The same dry weight of cells was used for the analysis. Student's *t*-test was done to test for statistical significance. n.s. not significant * = ≤0.05, ** = ≤0.01, *** = ≤0.001.

addition to the full length band [39] (Fig 1C). Thus, nickel, like fluconazole, activates the cleavage of Sre1. Collectively, these results demonstrate that activated Sre1 is critical for *C. neoformans* to tolerate Ni.

We hypothesized that Ni may outcompete other important metal cofactors (e.g., copper, iron, or zinc) in the *sre1Δ* mutant, rendering the strain unable to grow in the presence of Ni.

Typically, enzymes utilize specific metal cofactors for their activity and strict homeostasis mechanisms ensure correct metalation of proteins. When metal homeostasis becomes imbalanced, mis-metalation can occur, inhibiting protein function [46]. However, the addition of iron (Fe), copper (Cu), or zinc (Zn) to the Ni medium failed to rescue the growth defect of the *sre1Δ* mutant (Fig 1D). Mutants defective in iron and copper-specific regulators Cir1 [47] and Mac1 (also known as Cuf1) [48] did not show any growth defect in Ni-supplemented medium (S2A Fig), indicating that Sre1 plays a specific role in regulating cryptococcal tolerance to Ni.

We considered the possibility that poor growth of *sre1Δ* on Ni could indicate toxicity caused by accumulation of excessive intracellular Ni. To test this possibility, we analyzed the cellular Ni concentrations in wildtype (WT) H99, the *sre1Δ*, the *cir1Δ*, and the *mac1Δ* strains grown on RPMI, RPMI+Ni, or RPMI+DMG (Ni chelator) by Inductively Coupled Plasma Mass Spectrometry (ICP-MS) using a previously established method [49]. We also included a known Ni transporter mutant, *nic1Δ* [35]. The level of Ni associated with *sre1Δ* cells was comparable to WT when cells were cultured in Ni-supplemented medium (Fig 1E). By contrast, *nic1Δ*, *cir1Δ* and *mac1Δ* showed altered cellular accumulation of Ni (Fig 1E), with *nic1Δ* and *mac1Δ* accumulating 1.7 and 2.4 times more Ni than the wild type, respectively. Surprisingly, the *sre1Δ* mutant accumulated significantly more Cu and Zn than WT when cultured on Ni-supplemented medium (S2B Fig). However, the *sre1Δ* mutant was not hypersensitive to either copper or zinc (Fig 1D). The increased amount of Ni in *nic1Δ* cells is surprising but consistent with previous findings suggesting that Nic1 is not the only transporter of Ni [35]. Given that increased accumulation of Ni in *nic1Δ* and *mac1Δ* did not cause growth defect on media with Ni and that the *sre1Δ* mutant did not accumulate more Ni than the wild type, these results indicate that the growth defect of the *sre1Δ* mutant is not due to the over-accumulation of intracellular Ni.

## Ni causes up-regulation of ergosterol biosynthesis pathway (EBP) genes

Addition of copper and iron cause dramatic changes to cryptococcal transcriptome [50, 51]. To investigate the transcriptomic response to Ni exposure and the role of Sre1, we conducted a comparative transcriptome analysis by RNA deep sequencing (RNA-seq). Wildtype and *sre1Δ* cells were grown on RPMI plates with or without supplementation of Ni (0.25mM) or DMG (4mM) at 37˚C. RPMI was chosen as growth medium since it is a defined medium and commonly used for testing antifungal or stress susceptibility. The cells were cultured for eight hours prior to RNA extraction and sequencing.

We found 87 genes in H99 were upregulated and 71 genes downregulated on Ni versus the RPMI control (S1 Table). Here we consider genes showing at least a 2-fold change ($|\log_2$ (Fold Change)$| \geq 1$) in transcript level differentially expressed genes (DEGs). Six of the 87 upregulated genes were EBP genes (namely *ERG10*, *ERG8*, *ERG25*, *ERG6*, *ERG2*, *ERG4*: red spots outside of the shaded area in Fig 2A) and *SRE1* was also significantly upregulated on Ni ($\log_2$(Fold Change) = 1.05). Ten additional EBP genes were considered modestly upregulated, with $\log_2$(Fold Change) values between 0.3 and 0.99 (red spots within the shaded area in Fig 2A and 2B). The six highly upregulated EBP DEGs were spread throughout the pathway, with two occurring in the mevalonate pathway and four in the late pathway (Fig 2C). Interestingly, although Ni was able to upregulate ergosterol biosynthesis genes, DMG did not cause any significant downregulation of these genes (S2 Table). We also noted that addition of Ni did not change the transcript levels of *URE1* and *URE7*.

Given that Sre1 is known to regulate transcription of EBP genes, the impact of Ni on transcription of these genes might be largely due to the response of Sre1 to this metal. To assess this hypothesis, we examined the transcriptome of *sre1Δ* grown on Ni and in hypoxia

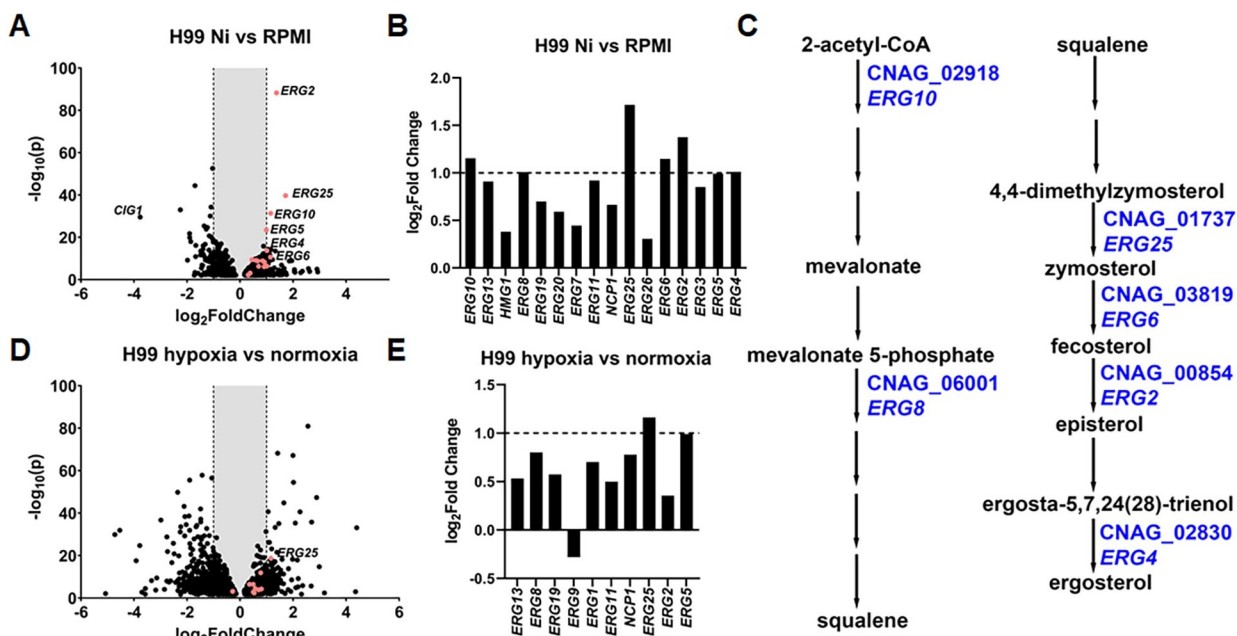

**Fig 2. Ni causes increased expression of multiple genes in the ergosterol biosynthesis pathway.** (**A**) Volcano plot of transcript changes of H99 growing on RPMI+Ni versus RPMI based on RNA-seq data. The red dots indicate EBP genes present in the data set. (**B**) Log$_2$(Fold Change) of EBP gene expression when grown on Ni. Differentially expressed genes are genes with a Log$_2$(Fold Change) value greater than 1, indicated by the dashed line. (**C**) Abbreviated ergosterol biosynthesis pathway. EBP genes differentially expressed on Ni were labeled in blue. (**D**) Volcano plot of transcript changes of H99 growing on RPMI+hypoxia versus RPMI+normoxia based on RNA-seq data. The red dots indicate EBP genes present in the data set. (**E**) Log$_2$(Fold Change) of EBP genes expression in hypoxia conditions.

conditions. When compared to the wild type grown on Ni, there were 12 negative DEGs that belonged to the EBP pathway (S4 Table and S3A Fig). 20 EBP genes showed statistically significantly altered expression although some did not reach the 2-fold cutoff. In comparison, hypoxia had a more limited impact on EBP gene expression in the *sre1Δ* mutant compared to the wild type (S5 Table and S3B Fig). Only four EBP genes (*ERG2*, *ERG5*, *ERG25*, and *ERG3*) were differentially expressed. Seven EBP genes showed altered gene expression. Thus, Ni exacerbates the reduced transcription of EBP genes in the *sre1Δ*.

Sre1 is also known to regulate the expression of Fe homeostasis genes [37]. Interestingly, Fe-starvation-related genes were overrepresented in the downregulated DEGs on Ni in wild-type cells. This is in contrast to EBP genes, which were upregulated in response to Ni. The most downregulated gene is an iron-starvation-responsive mannoprotein, *CIG1* [52]. Two oxidoreductases, *FRE7* and CNAG_02839, and two siderophore transporters, *STR1* and *SIT2*, were among the 20 most downregulated genes (S1 Table). All of these genes were significantly upregulated in cells grown on DMG-supplemented medium (S2 Table). Interestingly, addition of Ni to the RPMI medium decreased intracellular iron content in wild type but increased iron level in *sre1Δ* cells based on our ICP-MS data (S4A Fig). We speculated that Ni supplementation might have caused the *sre1Δ* cells to mistakenly sense that they were in an iron excess environment and that downregulation of iron-limitation response genes might have contributed to the hyper-sensitivity of *sre1Δ* to Ni. However, the observation that adding iron to the Ni supplemented medium did not restore growth of the *sre1Δ* mutant (Fig 1D) and that neither deletion nor overexpression of *CIG1* had any effect on the susceptibility of *sre1Δ* to Ni (S4B Fig) suggest that the ability to respond to intracellular Fe may not be the driving force behind the Ni sensitivity of *sre1Δ*.

As Sre1 is critical for hypoxia growth [53], and other divalent cations such as cobalt chloride ($CoCl_2$) have been characterized as hypoxia-mimicking agents [39,54], it is possible that Ni has a similar hypoxia-mimicking effect on *C. neoformans*. We found that $CoCl_2$ is highly toxic to *Cryptococcus* as both the wildtype and the *sre1Δ* cells were unable to tolerate 250μM $CoCl_2$ in RPMI media (S5A Fig). By contrast, wildtype cells grew readily on RPMI supplemented with Ni at the same concentration. Another transcription factor, Pas2, is important for remodeling cellular metabolism in response to hypoxia [53]. Deletion of this gene renders cells sensitive to hypoxia and $CoCl_2$ stress [53]. We hypothesized that if Ni mimics hypoxia, then *pas2Δ* would be sensitive to Ni as well. Indeed, *pas2Δ* was slightly more sensitive to Ni than WT, but much more tolerant than the *sre1Δ* mutant (S5B Fig). This is consistent with previous findings in which *pas2Δ* is less sensitive to hypoxia than is *sre1Δ*. In all, this data suggests that Ni and hypoxia have some shared effects on cryptococcal cells.

To further compare cryptococcal response to hypoxia and Ni, we performed comparative analyses of RNA-seq data between cells exposed to hypoxia and cells exposed to normoxia. Under hypoxia conditions (0.1% $O_2$, 5% $CO_2$, 37˚C), we found 167 upregulated DEGs and 434 downregulated DEGs (S3 Table), showing that hypoxia has a broader effect on gene expression than Ni. Despite the larger number of DEGs, hypoxia resulted in only one EBP gene, *ERG25*, being an DEG with more than a 2-fold change in transcript level (Fig 3D and 3E). The data indicates that Ni, relative to hypoxia, has a more narrow but profound impact specifically on the EBP. We also examined genes that were shared between wildtype cells grown exposed to Ni and wildtype cells exposed to hypoxia (S6 Table). We discovered that only 8 genes were upregulated and 28 genes were downregulated DEGs in both conditions (S3C Fig). Thus, Ni and hypoxia have distinct effects on cryptococcal transcriptome.

## Overexpression of *ERG25*, but not other four *ERG* genes tested, confers Ni tolerance

Sre1 is known to regulate ergosterol biosynthesis, and our RNA-seq data showed that Ni causes an upregulation of multiple EBP genes. We postulated that ergosterol deficiency due to the *SRE1* deletion may have contributed to *sre1Δ* Ni hypersensitivity. If this is true, then perturbations of the EBP pathway may also alter cryptococcal tolerance to Ni. So, we tested the deletion mutants of the non-essential *ERG3* and *ERG4* genes. Previous studies have shown that mutation of these genes perturbs ergosterol biosynthesis. For example, deletion of *ERG3* causes increased resistance to azoles (target Erg11) and Amphotericin B (bind to ergosterol) in *C. neoformans* and in *Candida* species [55–57]. Deletion of *ERG4* causes an increased sensitivity to caspofungin that targets β1–3 glucan synthase in the membrane [58]. *erg3Δ* and *erg4Δ* mutants grew like the wild type on RPMI+Ni (S6 Fig). Thus, it appears that perturbation of the EBP pathway in general does not alter cryptococcal tolerance to Ni.

To further interrogate our hypothesis, we decided to examine the impact of overexpression of ERG genes on Ni tolerance. We chose to overexpress ERG genes because most ERG genes are essential and cannot be deleted. Overexpression of ERG genes has been adopted as an effective approach to study the EBP pathway in *S. cerevisiae* [59]. To that end, we selected and overexpressed five EBP genes −*ERG2*, *ERG11*, *ERG25*, *ERG26*, and *ERG27* − in the wildtype and the *sre1Δ* backgrounds. To determine if these overexpressed *ERG* genes are functional, we first tested the susceptibility of these *ERG* gene overexpression strains in H99 background to fluconazole [60]. Fluconazole is an antifungal drug that inhibits Erg11, causing a reduction of ergosterol and a buildup of methylated sterols, collectively disrupting membrane stability [61]. We found that overexpression of any of the five EBP genes enhanced resistance to fluconazole, albeit at varied degrees (Fig 3A). As expected, overexpression of *ERG11*, the direct target of

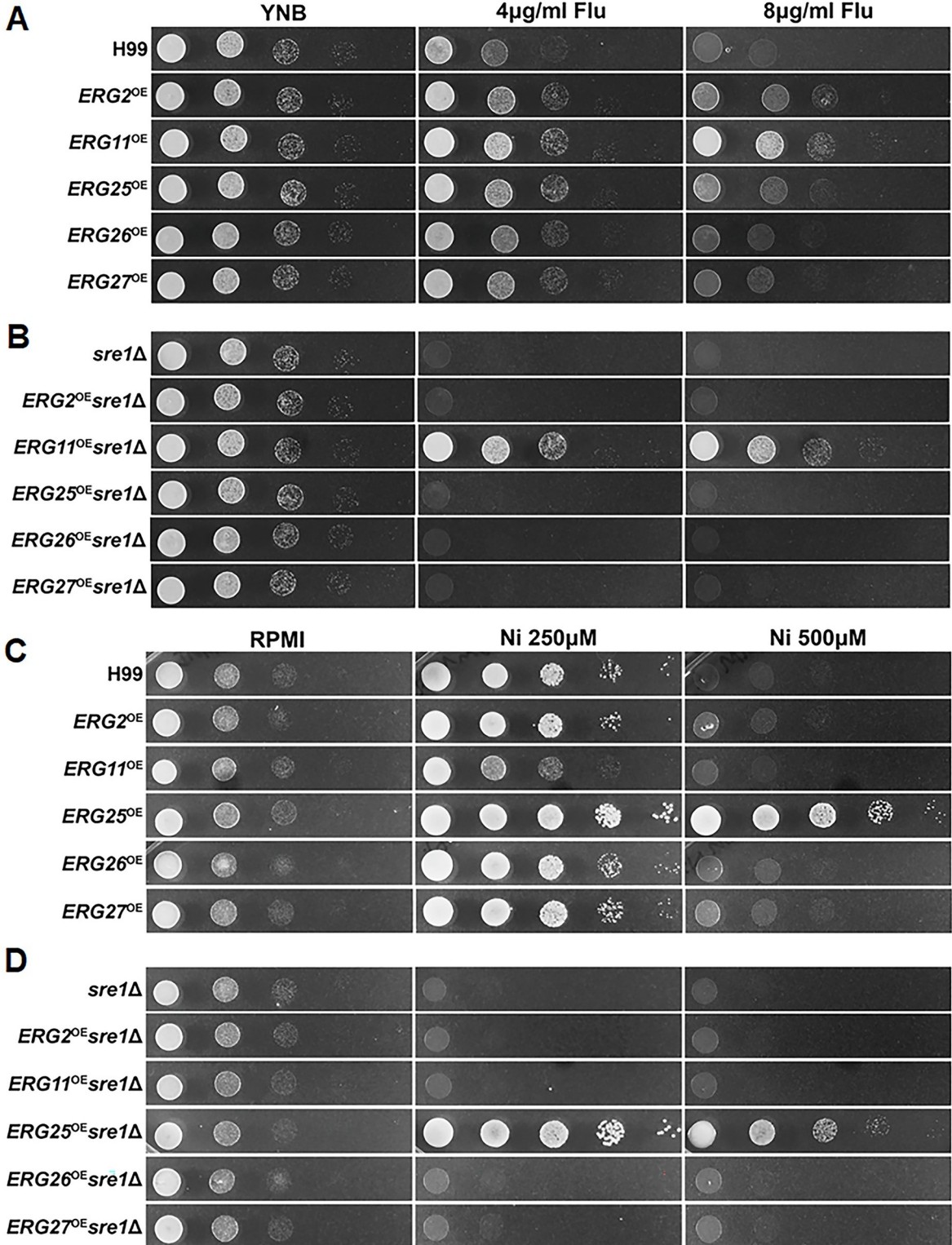

**Fig 3. Overexpression of *ERG25*, but not other *ERG* genes tested, drastically increases Ni tolerance in both wild type and *sre1Δ*. (A)** H99 strains with overexpression of the indicated *ERG* genes (*ERG2*, *11*, *25*, *26*, and *27*) were serially diluted and plated on RPMI plates with fluconazole (Flu) at 4 or 8 μg/ml. **(B)** The *sre1Δ* strains with overexpression of the indicated *ERG* genes (*ERG2*, *11*, *25*, *26*, and *27*) were serially diluted and plated on RPMI plates with Flu at the indicated concentrations. **(C)** The same strains as in Panel A were spotted onto RPMI media with the indicated concentrations of Ni. **(D)** The same strains as in Panel B were spotted onto RPMI media with the indicated concentrations of Ni. All plates were incubated for two days prior to imaging.

azole drugs, offered the highest level of resistance to fluconazole relative to the other *ERG* genes (Fig 3A). This result indicates that the overexpressed *ERG* genes are functional. However, when introduced to the *sre1Δ* mutant, none of the ERG overexpression was able to restore tolerance to fluconazole, with *ERG11* being the only exception (Fig 3B). This result reaffirms that Erg11 is the direct target of fluconazole and that overexpression of *ERG11* confers resistance to fluconazole regardless of the strain background.

After confirmation that these overexpressed EBP genes are functional, we sought to determine the effect of overexpression of these EBP genes on Ni tolerance. In contrast to what we observed in fluconazole resistance, the *ERG25*$^{OE}$ strain in wildtype background was exceedingly tolerant of Ni. The *ERG25*$^{OE}$ strain grew much better than the wild type on RPMI +500μM Ni, a condition that all other strains, including WT, were unable to tolerate (Fig 3C). Overexpression of *ERG25*, a known direct target of Sre1 [39], conferred marked Ni tolerance to the *sre1Δ* mutant as well. The *ERG25*$^{OE}$ *sre1Δ* strain was much more tolerant to Ni than even the wildtype strain (Fig 3D). Interestingly, overexpression of any of the other ergosterol biosynthesis genes, including *ERG11* and *ERG2* that lie upstream and downstream of *ERG25* respectively, did not confer Ni tolerance to either the wild type or the *sre1Δ* mutant. The overexpression of *ERG26* and *ERG27*, which encode enzymes that complex with Erg25, did not impact Ni tolerance in either strain backgrounds. This suggests that Erg25 specifically, not the EBP in general, plays a major role in mediating cryptococcal tolerance to Ni.

As we noted previously, hypoxia and Ni elicit both shared and distinct transcriptome changes in *Cryptococcus*, with Ni eliciting more profound and specific changes in the EBP pathway. *ERG25* is the shared gene upregulated by both stressors. A previous study identified *ERG25* as a multicopy suppressor of *scp1Δ* and *sre1Δ* sensitivity on CoCl$_2$, a known hypoxia-mimicking agent [39]. We found that overexpression of *ERG25* conferred tolerance to CoCl$_2$ in both wildtype and *sre1Δ* backgrounds (S7A and S7B Fig), in agreement with the previous study. Overexpression of other *ERG* genes tested failed to confer significant tolerance to CoCl$_2$ in either the wildtype or the *sre1Δ* background. When the strains were grown in hypoxia conditions, we found that all overexpression strains in the wildtype background grew similarly to the control (S7C Fig). However, the overexpression of *ERG25*, *ERG2*, *ERG11*, and *ERG26* all rescued the *sre1Δ* hypoxia growth defect albeit in varying degrees in that order (S7D Fig). The *ERG25* overexpression best rescued *sre1Δ* growth in hypoxia, which could be attributable to the upregulation of *ERG25* in the hypoxia condition that was indicated by our RNA-seq data (Fig 2D). The results support that upregulation of the EBP pathway genes generally enhances growth in hypoxia and confers resistance to fluconazole, but *ERG25* is specifically required for cryptococcal tolerance to cobalt and Ni.

## Ni alters the cellular lipid profile

Our results above demonstrate that Ni increases transcription of ergosterol genes and overexpression of *ERG25* in particular increases cryptococcal tolerance of Ni. To examine if Ni indeed impacts ergosterol levels, we extracted cellular ergosterol from wildtype and *sre1Δ* cells cultured on RPMI medium with or without the addition of Ni, and quantified ergosterol levels by measuring the absorbance at 282nm [62]. We found that the ergosterol content in wild type was reduced by 10% when exposed to Ni (Fig 4A). As expected, the ergosterol level in *sre1Δ* cells was lower under the normal growth condition with 80% of that in wildtype cells, and exposure to Ni caused a further reduction to 65% of that in untreated H99 cells. When exposed to Ni, the resulting amount of ergosterol in *sre1Δ* cells was similar to that in H99 cells exposed to fluconazole (Fig 4A). Filipin staining of ergosterol present in the outer leaflet of plasma membrane [60,63,64] showed that Ni exposure caused a 50% reduction in plasma membrane

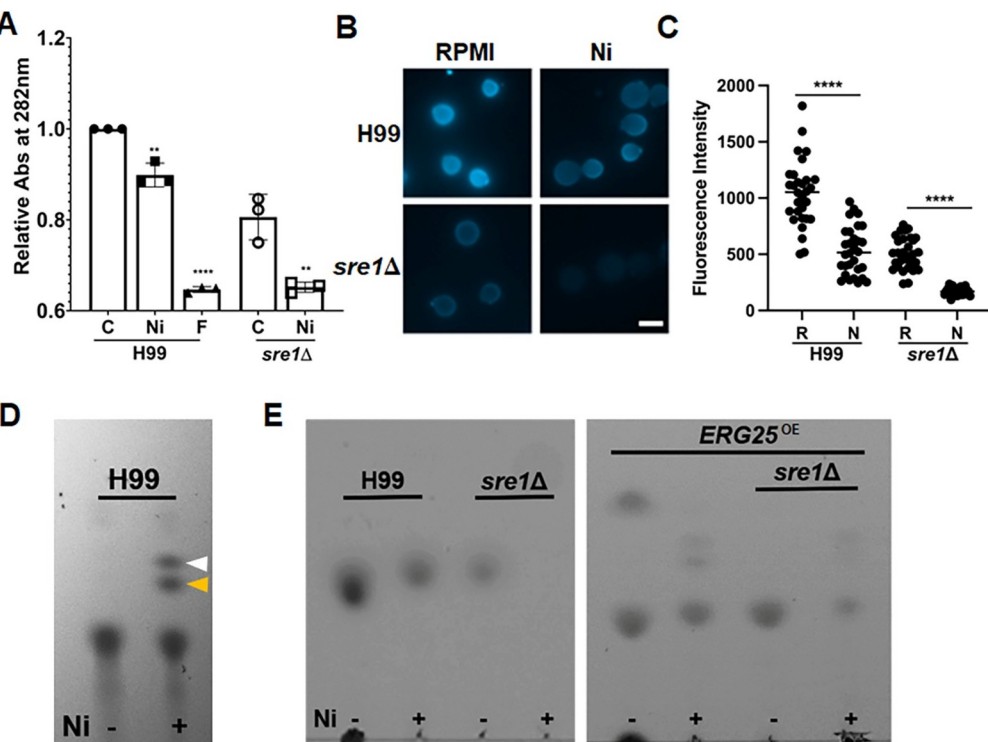

**Fig 4. Ni causes alteration of sterol profiles. (A)** Ergosterol was extracted from the indicated strains grown on RPMI (C), RPMI+250μM Ni (Ni), or RPMI+4μg/mL Flu (F). The extract was measured at 282nm. Student's t-test was used to assess statistical significance. ** = ≤0.01, **** = ≤0.0001 **(B)** H99 and *sre1*Δ cells were grown overnight on RPMI or RPMI+250μM Ni plates. Cells were harvested and incubated for 45 minutes in 25μM filipin III at 21°C in the dark. Cells were imaged with a Zeiss Imager M2 microscope. Scale bar = 5μm. **(C)** Quantification of fluorescence intensities of cells prepared as in panel B. RPMI (R), RPMI+ 250μM Ni (N). Mann-Whitney test was used to assess statistical significance **** = ≤0.0001. **(D)** Equal dry weight of wildtype cells grown on RPMI media alone (-) or RPMI+Ni (+) media were used to extract membrane sterols from indicated strains. 5μL of ergosterol extract were spotted onto glass backed HPTLC Silica gel plates. The white arrow indicates free fatty acids band. The yellow arrow indicates the methyl sterol band. The bottom band indicates ergosterol. **(E)** Lipid extractions from the indicated strains were spotted onto a TLC plate. 5μL of ergosterol extract was spotted onto glass backed HPTLC Silica gel plates as in panel D.

ergosterol in H99 based on fluorescence intensity (Fig 4B and 4C), consistent with the lower ergosterol levels in the presence of nickel measured spectrophotometrically (Fig 4B and 4C). The fluorescence intensity of *sre1*Δ cells was 50% of the wildtype level. Ni exposure reduced the fluorescence intensity even further to about 16% of that in H99 cells grown on RPMI (Fig 4C). Although both measurements revealed the same trend, the stronger reduction caused by the *SRE1* deletion or by Ni treatment measured by fluorescence intensity of filipin staining compared to spectrometry may be due to the fact that the extracts contain other lipids in addition to ergosterol or its intermediates.

Thin layer chromatography (TLC) analysis also revealed altered lipid profiles when cells were treated with Ni (Fig 4D). Based on previous literature using the same procedures for extraction and TLC [65], Ni causes an increase in the intensity of two of these bands, likely the methyl sterol (white arrow) and free fatty acid band (yellow arrow), and there is a slight decrease in the ergosterol band at the bottom (Fig 4D). Previous studies in yeast have shown that mutations that reduce the activity of *ERG25* caused a similar increase in the intensity of the methyl sterol and free fatty acid bands [65]. This result corroborates our hypothesis that Ni primarily targets Erg25 in the EBP pathway. In agreement with other measurements, the TLC analysis also revealed that ergosterol level was reduced in the *sre1*Δ mutant. The thick

ergosterol band was reduced compared to the wildtype strain when grown on RPMI medium, and this band was almost undetectable when the mutant was grown on RPMI with Ni (Fig 4E). The *ERG25* overexpression strain still showed a decrease in ergosterol content when plated on Ni media (Fig 4D). However, when *ERG25* was overexpressed in the *sre1Δ* background, an ergosterol band is still observable in contrast to the *sre1Δ* strain on Ni.

### Two histidine residues in Erg25 metal binding motifs are important for Ni tolerance

The results presented earlier indicate that Erg25 is critical for Ni tolerance in *C. neoformans*. Erg25 contains four conserved metal binding motifs enriched in histidine (S8 Fig). According to AlphaFold, several histidine residues are predicted to associate in a histidine-rich pocket present in Erg25 [66,67]. These histidine residues are from three of the four metal binding motifs and are not simply histidine residues proximal to each other in the primary sequence. In these regions, pairs of His residues (H187 and H272) are predicted to interact via cation-pi interactions (Fig 5A). I-TASSER, a protein structure prediction software [68–70], similarly predicts that these histidine residues are capable of interacting with a cation. We postulate that Ni binds to Erg25, and these His residues are critical for the function of Erg25.

To test our hypothesis, we mutated histidine residues 187 and 272, and overexpressed the *ERG25*[H187A H272A] allele in both the wild type and the *sre1Δ* mutant. Although overexpression

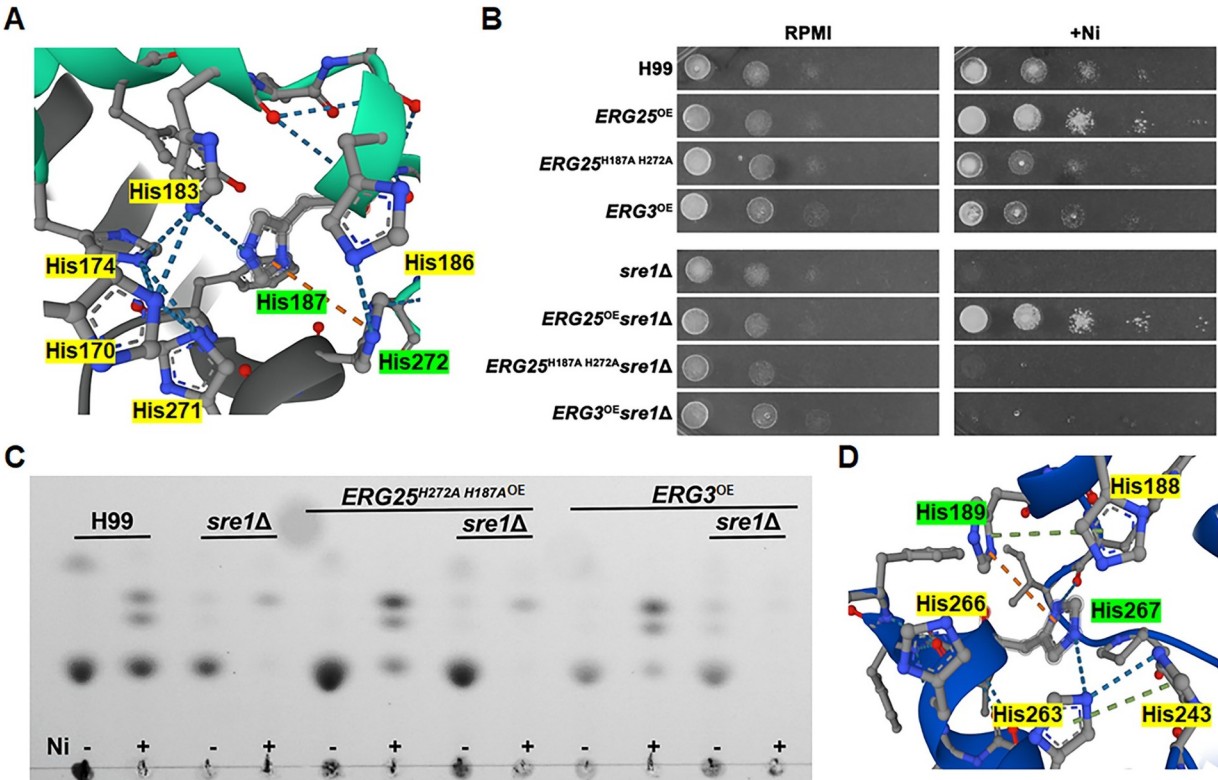

**Fig 5. Mutating histidine enriched pocket abolishes the ability of Erg25 to confer Ni tolerance. (A)** Alphafold generated images of histidine enriched pocket of Erg25. Predicted cation-pi interaction is indicated by an orange dashed line. **(B)** The indicated strains were serially diluted and spotted onto RPMI and RPMI+ 250μM Ni plates. The plates were incubated for 2 days before imaging. **(C)** Equal dry weight of cells was used to extract membrane sterols from indicated strains. 5μL of ergosterol extract were spotted onto glass backed HPTLC Silica gel plates. **(D)** Alphafold generated image of Erg3 histidine enriched pocket.

of the $ERG25^{H187A\ H272A}$ allele in wild type did not have any effect on growth on RPMI medium, it slightly reduced cryptococcal tolerance to Ni, in contrast to the much-enhanced Ni tolerance by the overexpression of the $ERG25$ allele (Fig 5B). Accordingly, the overexpression of the $ERG25^{H187A\ H272A}$ allele failed to rescue the growth defect of $sre1\Delta$ on Ni supplemented medium (Fig 5B). We speculate that mutations of these histidine residues prevented the binding of Erg25$^{H187A\ H272A}$ to Ni, which might have allowed Ni to bind to the native Erg25, compromising ergosterol biosynthesis. Moreover, nonfunctional Erg25$^{H187A\ H272A}$ may compete with the native Erg25 to complex with Erg26 and Erg27, further impairing the EBP pathway, rendering cells hypersensitive to Ni. Indeed, TLC analysis revealed that the $ERG25^{H187A\ H272A}$ overexpression strain had a larger reduction in ergosterol when exposed to Ni compared to the wild type (Fig 5D).

## Overexpression of *ERG3* or *URE7* does not rescue *sre1Δ* growth on Ni

We hypothesize that either Erg25 enzymatic function is required for nickel tolerance or Erg25 acts as a nickel sink because of its metal binding pocket can bind to nickel efficiently. Deletion of *SRE1* reduces Erg25 abundance and thus renders the fungus sensitive to nickel. Conversely, over-production of Erg25 confers *Cryptococcus* nickel resistance. However, these observations do not distinguish the two aforementioned hypotheses. To that end, we decided to overexpress another protein with a similar metal binding pocket. We expect that if the "nickel sink" hypothesis is true then overexpression of this protein would also confer nickel tolerance. Erg3 possesses similar conserved metal binding motifs as Erg25 (Fig 5D) [66,67]. *ERG3* is not on our DEG list because its transcript level increase in response to Ni was below 2-fold (Fig 2B). However, we found that unlike *ERG25*, overexpression of *ERG3* did not have any obvious impact on cryptococcal growth or Ni tolerance in either the wild type or the *sre1Δ* mutant (Fig 5B). Consistent with *ERG3* being non-consequential in conferring Ni tolerance, TLC analysis revealed a similar trend noted above regarding lipid changes in response to Ni in both wild type and in *sre1Δ* with or without *ERG3* overexpression. Therefore, Erg3 simply possessing a similar binding pocket to Erg25 is not sufficient for *sre1Δ* growth rescue on Ni.

We decided to test our hypotheses further with a known nickel-binding protein. Ure7 binds Ni as a chaperone protein during the activation of Ure1 [35]. Again, if the "nickel sink" hypothesis is true, we expect *URE7* overexpression would allow for Ni tolerance of the *sre1Δ* mutant. We first confirmed that our overexpression construct indeed increased the *URE7* transcript level in the wild type and we then deleted *SRE1* in the *URE7^OE* overexpression strain. We confirmed increased *URE7* transcript level in both WT and *sre1Δ* background via RT-PCR (S9 Fig). The *URE7* overexpression construct, once introduced into the *ure7Δ* mutant, was able to restore its urease activity assay based on CUA assay (Fig 6A), thus confirming that the overexpressed Ure7 is functional. Upon spotting strains onto RPMI media +/- 250μM Ni, we found that the *URE7* overexpression failed to restore growth of the *sre1Δ* mutant on Ni (Fig 6B). In all, the findings indicate that in contrast to *ERG25*, overexpression of *ERG3* or *URE7* does not confer nickel tolerance. Thus, Erg25 is likely not acting simply as a Ni sink.

## Ni reduces cholesterol in mammalian cells

Ni is known to affect lipid profiles in mammals. In humans and animals, exposure to Ni is associated with decreased serum cholesterol levels [71,72]. Given our finding about the impact of Ni on the sterol biosynthetic pathway in *C. neoformans* and the fact that the sterol biosynthetic pathway is highly conserved between fungi and humans/animals [73], we hypothesized that exposure of Ni will reduce cholesterol at the cellular level in mammals as well. Here, we cultured human epithelial A549 cells with or without addition of Ni at 10, 50 or 100 μM. We

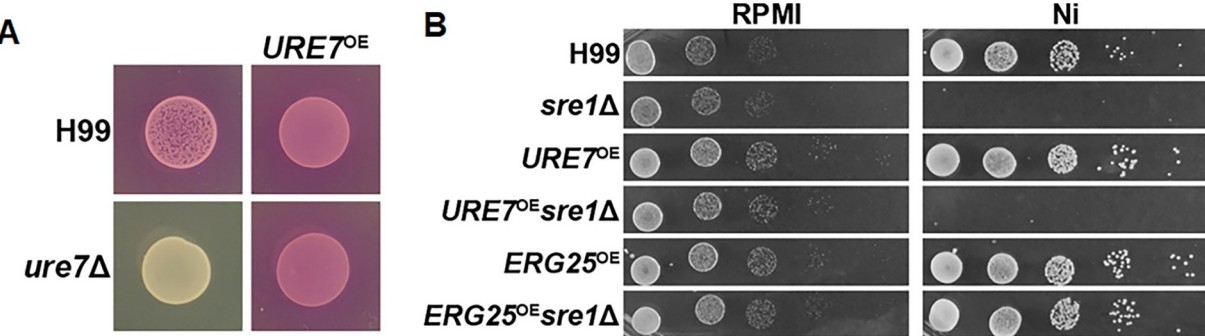

**Fig 6. Overexpression of *URE7* does not confer Ni tolerance to *sre1Δ*. (A)** H99, *ure7Δ*, and *URE7* overexpression in both strain backgrounds with cell density $OD_{600}$ = 3 were spotted onto CUA plates. The plates were incubated for three days and imaged. Urease activity is indicated by the yellow to pink color change. **(B)** The indicated strains were serially diluted and spotted onto RPMI and RPMI+ 250μM Ni plates. The plates were incubated for 2 days before imaging.

then stained the cells at 72 hours with filipin. The epithelial cells were able to grow desmosomes in all Ni concentrations tested (Fig 7A). We found that after passaging epithelial cells for 72 hours with 10, 50, or 100μM Ni, cells showed a 2.1, 4, or 5-fold reduction in filipin fluorescence intensity compared to unexposed cells (Fig 7A and 7B). This result suggests that Ni also reduces the amount of cholesterol in mammalian cells, similar to its ergosterol reduction effect in fungal cells. Whether Erg25 homolog Fet6 is also involved in nickel tolerance in mammals is yet to be tested.

## Discussion

Ni at high concentrations is toxic to cells, prokaryotic or eukaryotic [2,74,75]. Some bacteria species utilize Ni for essential protein functions but mammals are not known to require nickel for enzymatic functions [76–78]. Only one protein in *C. neoformans* (and in some other fungi), urease, is known to bind Ni. We have shown here that urease activity cannot occur when Ni is unavailable but urease is not required for the fungus to tolerate Ni. What mechanisms cryptococcal cells employ to regulate Ni homeostasis, particularly tolerance to Ni, remained unknown.

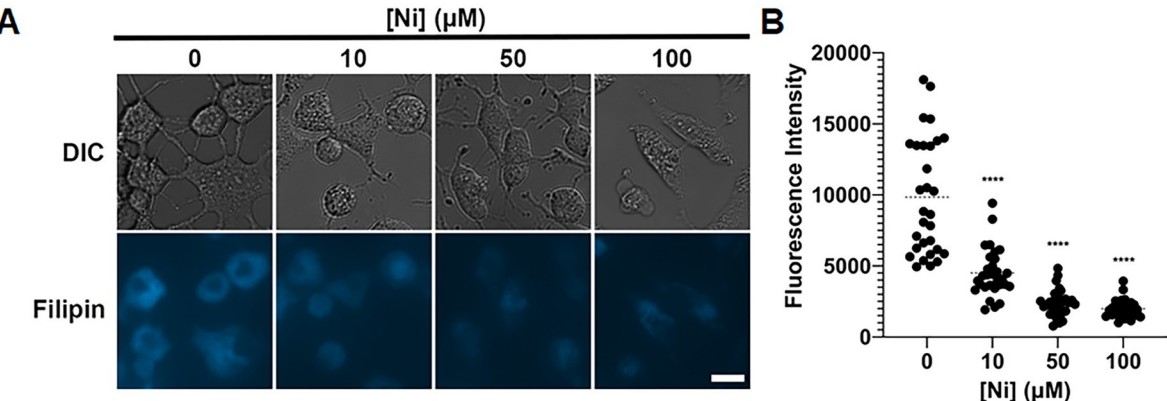

**Fig 7. Ni causes a decrease in membrane sterols in A549 lung epithelial cells. (A)** A549 lung epithelial cells were seeded into a 24-well glass bottom multi-well plate at $5 \times 10^4$ cells per well. Ni was added to the DMEM+FBS media to achieve the indicated concentrations and the cells were incubated at 37°C with 5% $CO_2$ for 72 hours. After being washed with warm PBS, the cells were incubated with Filipin III for 45 minutes and imaged. Scale bar = 15μm. **(B)** Fluorescence intensity was calculated via Zen Pro software. Mann-Whitney test was used to determine statistical significance. * = ≤0.05.

Ni has been shown to decrease plasma lipids; including total cholesterol, high-density lipoprotein-cholesterol, low-density lipoprotein-cholesterol (LDL) in a general population captured by NHANES study [72]. Ni is also capable of causing lipid peroxidation [11]. In rats, Ni depletion has been shown to increase the amount of cholesterol and LDLs [79]. However, in chickens, Ni supplementation did not alter cholesterol content [80]. We found that Ni supplementation decreases ergosterol content in *C. neoformans* and cholesterol in A549 human lung epithelial cells. Via RNA-sequencing, we have also characterized the role of Ni as a specific stimulant of the EBP pathway in the fungus.

Strikingly, Erg25 is the most upregulated EBP in response to Ni from these experiments and the only *ERG* gene that confers remarkably tolerance to nickel when overexpressed. This is in contrast to tolerance of fluconazole, where overexpression of multiple *ERG* genes conferred resistance even though the direct target Erg11 was the most effective. Erg25 is known to complex with Erg26 and Erg27 [59,81], but the overexpression of *ERG26* or *ERG27* does not have any significant impact on Ni tolerance. The stoichiometry of Erg25, Erg26 and Erg27 protein levels in the cell is not 1:1:1. In the fission yeast *Schizosaccharomyces pombe*, cells contain approximately 41 *ERG25*, 6 *ERG26*, and 3 *ERG27* RNA molecules per cell [82]. Erg25 in *S. pombe* was 2.5 times more abundant at the protein level than other complex members [83]. A similar phenomenon was observed in *S. cerevisiae* [84]. If Erg25 formed dimers or trimers, that may explain why the cell would produce Erg25 in such excess compared to other complex members. However, this has not been shown to be the case [81]. It is possible that Erg25 may perform a function outside of the studied complex that has not yet been revealed. Alternatively, Erg25 may control the rate limiting step and more Erg25 proteins are needed to produce the precursors used by the downstream enzymes. This is possible as the complex formation is likely more important for metabolic channeling rather than for specific enzymatic reactions carried out by these proteins *per se*, similar to subcellular compartmentalization of fungal secondary metabolism [85]. We suspect that Erg25 is able to serve as a chelator of Ni, and due to its abundance in the overexpression strains, still has sufficient level of the proteins to serve its function in the Erg25/Erg26/Erg27 complex in the sterol biosynthetic pathway in the presence of Ni. This ultimately allows the *sre1Δ* mutant to grow on Ni supplemented medium. Production of abundant Erg25$^{H187A\ H272A}$ could compete and thus compromise the activity of the native Erg25 protein, becoming a poison subunit in the complex. Its ill effect becomes apparent in the presence of Ni.

We argue that Erg25 does not simply act as a sink for intracellular Ni. The adverse effect of Ni on growth in *sre1Δ* is more likely attributable to its impact on Erg25's specific function, which cannot be recapitulated by overexpression of another enzyme with the same or similar metal binding pocket. Consistent with this idea, overexpression of *ERG3* is incapable of rescuing the growth of *sre1Δ* on Ni or conferring Ni tolerance to the wildtype cells despite the conserved histidine enriched region present in Erg3. That said, it is possible that this could be due to differences in protein structures that allow different levels of access to Ni ion. Ni-binding proteins are difficult to predict. Our screening of gene deletion mutants of genes that encode histidine enriched proteins did not identify any additional genes that are essential for Ni tolerance. We also used Ni-beads to try to identify proteins in *C. neoformans* that can bind Ni, and the only reliable nickel-binding protein pulled down in these assays was Ure7. The finding that overexpression of this known nickel-binding protein Ure7 does not rescue nickel sensitivity in the *sre1Δ* mutant further bolstered the conclusion that Erg25 is not acting simply as a nickel sink, but rather its function integrity is required for nickel tolerance.

In all, we have described that Ni impacts sterol profiles in *C. neoformans* and cells respond by upregulating the sterol biosynthesis pathway in order to tolerate this metal. Multiple lines of evidence support the idea that Ni specifically targets Erg25, analogous to how fluconazole

targets Erg11. Furthermore, the reduction of sterols in the presence of Ni is conserved in both fungi and mammals. Whether Ni acts on the same targets in fungi and mammals for such an effect is unknown. The current findings could stimulate and guide future investigation of effective ways to mitigate or prevent nickel toxicity.

## Materials and methods

### Strains and growth conditions

*C. neoformans* strains used in this study are listed in S7 Table. Strains were stored at -8˚C in 15% glycerol stocks and freshly streaked onto yeast peptone dextrose (YPD) media prior to experimentation. Cells were maintained on YPD medium at 30˚C unless stated otherwise. RPMI 1640 medium (catalog number. SH30011.04, Cytiva)+ 165mM MOPS was prepared and adjusted to pH = 7 for all experiments in which cells were grown on RPMI medium.

### Gene deletion mutant library screen for sensitivity to Ni or DMG

To identify genes deletion mutants sensitive to the Ni chelator DMG or Ni, the transcription factor and kinase deletion libraries generated by Dr. Yong-Sun Bahn's group [43,44], and the partial genome deletion library generated by Dr. Hiten Madhani's group were replicated into RPMI liquid media at 37˚C and incubated for 1–2 days to allow them to reach the stationary phase. The libraries were replicated in RPMI medium to ensure that any phenotype we observed on our RPMI+Ni or RPMI+DMG plates was not due to the mutant's growth impairment in RPMI media. These cultures were spotted onto solid RPMI supplemented with 4 mM DMG or 250 μM Ni and grown at 37˚C for three days to assess sensitivity visually.

### Metal assays

To examine sensitivity to various metals, strains were grown in YPD liquid medium overnight with shaking at 220 rpm at 30˚C. Cells were collected, washed once with sterile water, and adjusted to a cell density of $OD_{600} = 1$. The cells were then serially diluted in 10-fold and spotted onto RPMI plates with the indicated metals: $CuSO_4$ (Cu), $ZnCl_2$ (Zn), and $FeSO_4$ (Fe) at 250 μM. Unless otherwise indicated, RPMI+Ni plates contained 250 μM $NiSO_4$. The plates were incubated at 37˚C for two days before imaging.

### Urease activity

Urea Agar (Becton Dickinson 211795) was prepared following the manufacturer's protocol. Plates were supplemented with indicated concentrations of DMG. Cryptococcal cells of the indicated strains ($OD_{600} = 3$) were spotted onto agar plates and incubated at 30˚C for 2 days before imaging.

### Fluconazole sensitivity assay

To examine fluconazole sensitivity, similarly prepared cells with serial dilutions were spotted onto yeast nitrogen base (YNB) agar as well as YNB with fluconazole at the indicated concentrations.

### Western blot

Strain Linlab7787 expressing 3xFlag-Sre1 were cultured on RPMI, RPMI+ 250μM Ni, and RPMI+4μg/mL fluconazole plates. The cells were collected by centrifugation and the supernatant was discarded. The cell pellet was frozen in liquid nitrogen. 1mL pre-chilled 0.2 M NaOH

(0.2% β-ME) and 0.5mm glass beads were added to the cell pellet. The cells were disrupted at 4°C with a bead beater (Next Advance) 5 times at 1 min working and 1 min rest. The supernatant was transferred to a new Eppendorf tube and 75μL of 100% trichloroacetic acid (TCA) was added. After 10-minute incubation on ice, the extraction was centrifuged at 12,000rpm for 5 minutes at 4°C and the supernatant was discarded. The pellet was resuspended in 100μL 1M Tris and was denatured with an SDS-containing loading buffer prior to electrophoresis on an SDS-12% PAGE gel. Samples separated on the SDS-PAGE gel were transferred to polyvinylidene difluoride (PVDF) membrane (Millipore), using the eBlot L1 Fast Wet Protein Transfer System (GenScript) using preset protocols. The blots were incubated with Mouse anti-Flag antibody diluted 1:2,000 (Sigma; Lot #: SLCJ3741), washed, and then incubated with Rabbit anti-mouse secondary antibody diluted 1:20,000 (Clontech Inc.). Signals were detected using enhanced chemiluminescence (ECL) according to manufacturer instructions (Pierce). Protein loading was confirmed via Coomassie staining.

## Gene manipulation

All primers and plasmids used in this study are listed in S8 Table. For gene overexpression, open reading frames (ORFs) of the indicated genes were amplified by PCR from *C. neoformans* H99 genomic DNA and cloned into vectors containing *TEF1* (pLinlab995) or *GPD1* (pLinlab1059) promoters. Plasmids were confirmed via two rounds of restriction enzyme digestion. M13F and M13R primers were used to PCR amplify the donor DNA, which was introduced into the indicated recipient strains using the Transient CRISPR-Cas9 coupled with Electroporation (TRACE) protocol as we described previously [86]. Constructs were integrated into the safe haven *SH2* region [87].

For gene deletion of *SRE1*, the deletion cassette was amplified from genomic DNA (gDNA) of a *sre1Δ* mutant, which is part of the *C. neoformans* transcription factor deletion library generated by Dr. Yong-Sun Bahn and colleagues [44]. To generate the sgRNA for the *SRE1* deletion, the *U6* promoter was amplified from JEC21 gDNA and the sgRNA scaffold was amplified from plasmid pDD162, using primer pairs Linlab4627/Linlab7751 and Linlab4628/Linlab7752, respectively. The *U6* promoter and sgRNA scaffold pieces were fused together by overlap PCR with primers Linlab4594/Linlab4595 to generate the final sgRNA construct as described previously [86,88].

Overexpression and deletion constructs were transformed into the indicated *C. neoformans* strains via TRACE [86, 88]. Transformants were selected on YPD medium with 100 μg/ml of nourseothricin (NAT), 100 μg/ml of neomycin (NEO), or 200 μg/ml of hygromycin (HYG) depending on the drug marker used.

The successful deletion of *SRE1* was screened via diagnostic PCR. Primer pair Linlab4895 (a *SRE1* promoter forward) and Linlab3792 (a reverse primer inside the NAT cassette) were used to ensure that the drug marker was inserted into the *SRE1* locus. Primers Linlab8488 and Linlab4897, which both lie on the *SRE1* open reading frame, were used to ensure that the ORF was missing. The successful integration of EBP gene overexpression constructs into the *SH2* region was screened via 3-primer PCR [89]. Since the construct can insert into the *SH2* region in either the forward or the reverse direction, primer Linlab5936, a reverse primer on the overexpression construct was paired with *SH2* sequencing primers Linlab4814 and 4815. A band indicative of the direction the construct was inserted into the genome, or if it was not inserted would be amplified with three primer PCRs.

The *ERG25*$^{H189A\ H272A}$ overexpression strain was constructed by amplifying the gene with primers with nucleotide changes that would mutate the indicated histidine codons to alanine. The successfully amplified ORF was cloned into pLinlab995 to be under the control of the

constitutively active *TEF1* promoter. All overexpression constructs were PCR amplified with M13F and M13R, and transformed into the *SH2* region in the indicated recipient strains using TRACE.

## RNA Extraction and Real-Time PCR

Real-Time PCR (RT-PCR) was used to confirm the EBP gene and *URE7* overexpression strains. Cells were grown in YPD liquid cultures with shaking at 30°C overnight. Cells were collected, flash-frozen with liquid nitrogen, and lyophilized overnight. Desiccated cells were disrupted with glass beads, and total RNA was extracted using the PureLink RNA Mini Kit (Invitrogen) according to the manufacturer's instructions. To remove any potential DNA contamination, samples were treated with DNase using the TURBO DNA-free Kit (Invitrogen) following the manufacturer's protocol. First-strand cDNA was synthesized using the GoScript Reverse Transcription System (Promega) following the manufacturer's instructions. Power SYBR Green (Invitrogen) was used for all RT-PCR reactions. *TEF1* was used as an internal control for all RNA samples. All RT-PCR primers used are listed in S2 Table. Relative transcript level was determined using the ΔΔCt method as we described previously [90] and statistical significance was determined using Student's t-test.

## RNA Deep sequencing

For transcriptome analysis in response to DMG, Ni, or hypoxia, overnight cultures in liquid YPD of the indicated strains were collected and washed twice with sterile $dH_2O$. Approximately $3x10^8$ cells were plated onto RPMI plates with or without supplementation with 250μM Ni or 4mM DMG. Cells were then incubated for eight hours at 37°C. Under the hypoxia condition, cells were cultured on RPMI plates for eight hours in the hypoxia chamber set to 0.1% $O_2$, 5% $CO_2$. The hypoxic environment was maintained using a Biospherix C chamber with $O_2$ levels controlled by a Pro-Ox controller and $CO_2$ levels controlled by a Pro-$CO_2$ controller (Biospherix, Lacona, NY, USA). At the designated time, cells were scrapped from the plates quickly, snap-frozen with liquid nitrogen, and then lyophilized. Total RNA from these cell samples was extracted using the PureLink RNA mini kit (Life Technologies) as described earlier.

RNA samples were sent to GENEWIZ, Inc. for sequencing (polyadenylated RNA enrichment, non-strand-specific, paired end 150 bp on Illumina HiSeq platform). The raw reads were trimmed using Trim_Galore (0.6.5) and aligned to *Cryptococcus neoformans* var. *grubii* H99 reference genome using STAR (2.7.1a). The alignment files (Bam) were used to generate read counts and Fragments Per Kilobase of transcript per Million mapped reads (FPKM) with Cufflinks (2.2.1). Differential gene expression analysis was performed using DESeq2 with false discovery rate (FDR) adjusted p-value ≤ 0.05 as threshold.

## RNA-Seq Data availability

The raw sequencing reads from this study have been submitted to the NCBI Sequence Read Archive (BioProject PRJNA1082343).

## Microscopy

For fluorescence observation of filipin staining samples were examined under a Zeiss Imager M2 microscope equipped with an AxioCam 506 mono camera. Filipin was visualized with the FL Filter Set 49 DAPI (Carl Zeiss Microscopy). The fluorescence intensity was quantified via Zen Pro software (Carl Zeiss Microscopy).

## Inductively Coupled Plasma Mass Spectrometry (ICP-MS)

For metal accumulation analysis, 50 mL cultures of *C. neoformans* strains were grown on YPD for 16 hours at 30˚C. Cells were collected by centrifugation and washed twice with sterile $dH_2O$. $3x10^8$ cells suspended in water were plated onto RPMI, RPMI+ 250 µM Ni, and RPMI + 4 mM DMG. The plates were incubated at 37˚C overnight. Cells were harvested from the plates and washed twice with sterile $dH_2O$. Approximately 200mg of cell pellets were aliquoted into pre-weighed tubes. Most of the $dH_2O$ was removed and the cell pellet was heat-killed at 95˚C for 20 minutes. Heat-killed cell samples were submitted to the Center for Applied Isotope Studies Plasma Chemistry Laboratory at the University of Georgia for *ICP-MS*.

Metals analysis was performed by Dr. Sarah Jantzi at the Plasma Chemistry Laboratory, Center for Applied Isotope Studies, University of Georgia. Samples, reference material, and method blanks were digested in PTFE vessels (Savillex, USA) using 0.5 mL trace metal grade concentrated nitric acid (Fisher Scientific, USA) for 1 hour 95˚C, followed by 0.5 mL trace metal grade hydrogen peroxide (Fisher Scientific) for 1 hour at 95˚C. Digestates were diluted with deionized water to 2% w/w nitric acid and the concentrations of Fe, Co, Ni, Cu, and Zn were determined by inductively-coupled plasma mass spectrometry (ICP-MS) using an indium internal standard. A Thermo X-Series 2 ICP-MS with collision cell technology and chilled spray chamber (Thermo, Germany) was used in kinetic energy discrimination (KED) mode with 8% hydrogen in helium to reduce interferences.

## Ergosterol extraction

The same number of cells ($5x10^7$) for the indicated strains were plated onto RPMI, RPMI + 250 µM Ni and incubated at 37˚C for 2 days. The cells were collected with sterile water and were snap-frozen in liquid nitrogen. The cells were then lyophilized overnight. Samples were normalized to the same dry weight and desiccated cells were disrupted manually with glass beads. Ergosterol was extracted using a previously established protocol [62]. Briefly, 3mL 25% alcoholic KOH was added to the cells and transferred to a borosilicate glass tube with screw cap, vortexed for 1 minute, and incubated at 85˚C for 1 hour. After cooling to room temperature, 1 mL of sterile water and 2 mL of n-heptane were added to each tube. Each tube was vortexed for 3 minutes and the solution was allowed to separate for ~5 minutes without centrifugation. The n-heptane layer was transferred to a microcentrifuge tube and concentrated via speedvac. 30 µL of n-heptane was added to the concentrated pellet for TLC analysis.

## Thin Layer Chromatography (TLC)

A standard curve to quantify amounts of ergosterol was done by creating a 2 mg/mL stock of ergosterol in chloroform. This stock was serially diluted in 2-fold increments. 5 µL of each sample was spotted onto 10x20cm HPTLC Silica gel coated glass plates (EMD Chemicals) with 1µL drop at a time. The spots were placed 0.5 cm apart and 1 cm from the bottom of the plate. A mobile phase of petroleum ether, diethyl ether, and acetic acid (vol 85:15:1) was used to run the TLC plate in a covered TLC chamber. The TLC plate was then allowed to air dry in the chemical hood, and was developed for 45 minutes with iodine crystals in the covered TLC chamber and imaged. The density of the spots was quantified via the standard curve data.

## Mammalian Cell Culture with or without Ni

A549 cells are human type II lung epithelial cells. A549 cells were maintained in Dulbecco's modified Eagle's medium (DMEM) supplemented with 10% fetal bovine serum (FBS) and incubated at 37˚C with 5% $CO_2$ in 250mL CellStar culture flasks (Greiner Bio-One). To test

the effect of Ni on membrane lipids, A549 cells were seeded into 24-well glass bottom cell culture plates (Southern LabWare) at a concentration of $5 \times 10^4$ cells per well in DMEM+FBS media with or without Ni. Concentrations of Ni used were 0, 10, 50, and 100μM. After 72 hours of growth, cells were washed with warm sterile PBS, and then incubated in 25 μg/mL filipin III (Cayman Chemical) stain for 45 minutes at 22°C in the dark before imaging. The cells were viewed using a Zeiss Axio Observer 7 inverted microscope using a Plan-APOCHROME 20x objective lens (Carl Zeiss Microscopy).

## Supporting information

**S1 Fig. Urease is not required for Ni tolerance.** (A)H99 cells were serially diluted and spotted onto RPMI and RPMI+Ni at the indicatedconcentrations. The plates were imaged after two days of incubation. (B) The indicated strains with cell density OD600 = 3 were spotted onto Christensen Urea Agar (CUA) plates. The plates were incubated for three days and imaged. Urease activity is indicated by the yellow to pink color change of the media due to alkalization of the media by released ammonia. (C) Two molecules of DMG chelate one molecule of Ni. (D) H99 and ure1Δ with cell density OD600 = 3 were spotted onto Christensen Urea Agar (CUA) plates with increasing concentrations of DMG. The plates were incubated for three days and imaged. (E) The indicated strains were serially diluted and spotted onto RPMI and RPMI+ 250μM Ni plates. The plates were imaged after two days of incubation.
(PDF)

**S2 Fig. Metal accumulation does not dictate metal sensitivity.** (A) The indicated strains were plated onto RPMI media with or without Ni, and incubated at 37°C for 2 days. (B) ICP-MS quantification of Cu (left) and Zn (right) concentrations in cells grown on RPMI+Ni. The same dry weight of cells was used for the analysis. Student's t-test was used for statistical analysis. **: $p \leq 0.01$, ***: $p \leq 0.001$.
(PDF)

**S3 Fig. Ni and hypoxia have distinct impacts on transcriptome.** (A) Volcano plot of transcript changes of sre1Δ grown on RPMI+Ni versus H99 grown on RPMI+Ni based on RNA-seq data. (B) Volcano plot of transcript changes of sre1Δ grown on RPMI+hypoxia versus H99 grown on RPMI+hypoxia based on RNA-seq data. The red dots in both panels indicate EBP genes present in the data set. (C) Plot of transcript changes of genes shared between H99 Ni vs RPMI and H99 hypoxia vs normoxia data sets. Red highlighted genes are EBP genes. In all panels, dots that fall outside of the shaded grey areas are DEGs.
(PDF)

**S4 Fig. The *sre1Δ* sensitivity to Ni is not due to iron starvation.** (A) ICP-MS data showing intracellular iron concentrations in H99 and sre1Δ on RPMI (C), RPMI+250μM Ni (N), and RPMI+DMG (D). Student's t-test was performed to assess statistical significance. ** = $\leq 0.01$, ns = not significant. (B) The indicated strains were serially diluted and spotted onto RPMI and RPMI+250μM Ni media. Plates were incubated at 37°C for two days prior to imaging.
(PDF)

**S5 Fig. Ni and Cobalt (Co) elicit overlapping and different effects on growth.** (A) Wildtype H99 and sre1Δ cells were serially diluted and spotted onto RPMI, RPMI+250μM Ni, and RPMI+250μM Co. The plates were incubated at 37°C for two days prior to imaging. (B) H99, pas2Δ and sre1Δ cells were serially diluted and spotted onto RPMI, and RPMI with the indicated concentration of Ni. The plates were incubated at 37°C for two days prior to imaging.
(PDF)

**S6 Fig. Neither ERG3 nor ERG4 is required for cryptococcal tolerance of Ni.** Cells of the indicated strains were serially diluted and spotted on RPMI and RPMI+250µM Ni. Plates were incubated at 37˚C for two days prior to imaging.
(PDF)

**S7 Fig. *ERG25* is required for tolerance of Co and *ERG* overexpression can partially restore the growth defect of *sre1Δ* in hypoxia.** (A) Cells of H99 with overexpression of the indicated ERG genes were serially diluted and plated onto YNB plates with CoCl2 at the indicated concentration. (B) Cells of the sre1Δ mutant with overexpression of the indicated ERG genes were serially diluted and plated on RPMI plates with the indicated concentrations of CoCl2. (C) Cells of the same strains as in Panel A were spotted onto YPD media and incubated in ambient air (normoxia) or hypoxia (0.1% O2, 5% CO2) conditions. (D) Cells of the same strains as in Panel B were spotted onto RPMI media with the indicated concentrations of Ni. All plates were incubated for two days prior to imaging.
(PDF)

**S8 Fig. Erg25 protein sequence contains histidine residues in predicted metal binding regions.** C. neoformans Erg25 protein is 343 amino acids in length. Four histidine enriched putative metal binding motifs are boxed in various colors. The histidine residues predicted to interact with a cation are highlighted in yellow.
(PDF)

**S9 Fig. *URE7* is overexpressed in both wildtype and *sre1Δ* backgrounds.** RT-PCR data was generated by harvesting cells from overnight YPD cultures of (A) wild type and URE7OE in the wild type background as well as (B) sre1Δ and sre1ΔURE7OE. Housekeeping gene TEF1 was used as an internal control to ensure the quality of the original RNA sample used for cDNA amplification and for normalization. Student's t-test was used for statistical analysis. ***: $p \leq 0.001$, ****: $p < 0.0001$.
(PDF)

**S1 Table. Differentially expressed genes H99 Ni vs H99 RPMI.**
(XLSX)

**S2 Table. Differentially expressed genes H99 DMG vs H99 RPMI.**
(XLSX)

**S3 Table. Differentially expressed genes H99 Hypoxia vs H99 normoxia.**
(XLSX)

**S4 Table. Differentially expressed genes *sre1Δ* Ni vs H99 Ni.**
(XLSX)

**S5 Table. Differentially expressed genes *sre1Δ* Hypoxia vs H99 Hypoxia.**
(XLSX)

**S6 Table. Differentially expressed genes shared between H99 Ni vs H99 RPMI and H99 hypoxia vs H99 normoxia data sets.**
(XLSX)

**S7 Table. Fungal strains used in this study.**
(XLSX)

**S8 Table. Primers and plasmids used in this study.**
(XLSX)

## Acknowledgments

We thank all Lin lab members for their helpful suggestions. We also thank Dr. Sarah Jantzi at the Plasma Chemistry Laboratory, Center for Applied Isotope Studies, University of Georgia for her assistance and expertise with ICP-MS. We also thank the Yong-Sun Bahn laboratory for generating the transcription factor and kinase deletion collections, and the Madhani laboratory for generating the deletion collection (funded by NIH R01AI100272).

## Author Contributions

**Conceptualization:** Amber R. Matha, Robert J. Maier, Xiaorong Lin.

**Data curation:** Xiaofeng Xie.

**Formal analysis:** Amber R. Matha, Xiaofeng Xie, Xiaorong Lin.

**Funding acquisition:** Xiaorong Lin.

**Investigation:** Amber R. Matha, Xiaorong Lin.

**Methodology:** Amber R. Matha, Xiaofeng Xie, Xiaorong Lin.

**Project administration:** Xiaorong Lin.

**Resources:** Xiaorong Lin.

**Validation:** Amber R. Matha.

**Visualization:** Amber R. Matha.

**Writing – original draft:** Amber R. Matha.

**Writing – review & editing:** Amber R. Matha, Xiaofeng Xie, Robert J. Maier, Xiaorong Lin.

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
