## [Decision Letter · Decision Letter 0]

29 May 2024

Dear Xiaorong,

Thank you very much for submitting your Research Article entitled 'Nickel tolerance is channeled through C-4 methyl sterol oxidase Erg25 in the sterol biosynthesis pathway' to PLOS Genetics.

The manuscript has now been evaluated by two independent peer reviewers. As you will below, the reviewers differ in their opinions. Reviewer 2 in particular is not convinced that you have shown that nickel directly impacts Erg25 activity. However, there are several relatively straightforward analyses that could address some of their concerns. First, it is necessary to reconstitute the sre1 mutant to be sure that the phenotypes observed are due to the loss of Sre1. Second, you should confirm by direct measurement (e.g. RT-PCR) that overexpression of the target genes does in fact lead to increased expression. Third, the manuscript would be strengthened by increasing the analysis of the RNA-seq data sets. You have discussed the relationship between nickel and cobalt, but it is not clear whether you think that both are acting in the same way, i.e. as hypoxic mimics. In Cryptococcus and other fungi, the response to cobalt overlaps with hypoxia, but is not identical. You have suggested (mostly in supplementary material) that the response to nickel and cobalt is not the same. Can you expand on this, for example by comparing the transcriptional response to nickel and cobalt using published data? Do your his mutants in Sre1 affect sensitivity to cobalt? Reviewer 2 has suggested measuring Sre1 activation in response to nickel. This is not absolutely essential, but it would strengthen your hypothesis. There are some apparent contradictions that should be addressed in more detail, such as why a nic1 deletion results in increased nickel, and why overexpressing ERG11 results in increased sensitivity to nickel, and apparently to cobalt also (Fig. S6).

Based on the reviews, we will not be able to accept this version of the manuscript, but we would be willing to review a revised version. We cannot, of course, promise publication at that time.

If you decide to revise the manuscript for further consideration at PLOS Genetics, please aim to resubmit within the next 60 days, unless it will take extra time to address the concerns of the reviewers, in which case we would appreciate an expected resubmission date by email to plosgenetics@plos.org.

We are sorry that we cannot be more positive about your manuscript at this stage. Please do not hesitate to contact us if you have any concerns or questions.

Yours sincerely,

Geraldine Butler

Section Editor

PLOS Genetics

Geraldine Butler

Section Editor

PLOS Genetics

Reviewer's Responses to Questions

**Comments to the Authors:**

Reviewer #1: The authors explore mechanisms of nickel tolerance in the human pathogenic fungus Cryptococcus neoformans. Previous work established that urease enzymes in this microorganism require Ni, but little is known about the cellular response to excess Ni. Using an established collection of strains with mutations in transcription factors, the authors identified the sterol response pathway regulator Sre1 as required for growth in the presence of high concentrations of Ni. This is the major initial finding upon which the rest of the paper builds, and the authors explore various aspects of Sre1 biology and activity to find specific ways Ni is handled in the fungal cell. Fungal Sre1 proteins have been shown to be activated by conditions as distinct as hypoxia and pH, and metal bioavailability may be a common linking theme for Sre1 activation. Interestingly, Ni can also activate the mammalian HIF1 protein, another transcriptional regulator that similarly coordinates hypoxic cellular responses.

The authors’ finding is strengthened by their documentation that other proteins involved in Sre1 activation (Scp1, Stp1) are similarly required for Ni tolerance. The authors furthermore explore targets of Sre1, including genes in the ergosterol biosynthetic pathway. The main finding is that Erg25, when overexpressed, increased cryptococcal Ni tolerance, making it somewhat unique among other Erg proteins. Blocking the Er5 pathway itself does not significantly affect Ni tolerance. Also, mutating metal-binding histidine residues results in a poorly functioning protein with regards to Ni tolerance. The reasons for this are not completely clear since the native ERG25 gene is still present. However, the authors offer reasonable explanation in the text.

The authors include a very nice control for their ERG gene overexpression study by documenting increased fluconazole tolerance in these strains.

Minor points:

1) Line 72. I would reword this sentence to minimize the results placed in the introduction; e.g. “However, the role of urease in C.n. nickel tolerance is unknown” Or similar

2) Line 144. The strain name H99 is used extensively in the manuscript and Figures. It might be more appropriate to simply refer to this strain as “WT”, especially in figures.

3) Line 116. The Legend refers to “RPMI+Ni (Ni)” but the figure just has “N” and not “Ni”

4) Line 136. “Fig 1B” should be “Fig 1C”.

5) Line 145. I would remove phrases such as “To our surprise” and simply state the results.

6) Line 216. “… much more tolerance …”

7) Is it possible to demonstrate whether Sre1 activation (e.g. protein cleavage) is indeed induced by excess Ni?

8) Fig3. Why does Erg11 over-expression result in increased sensitivity to Ni?

9) Line 308. The authors link filipin straining intensity very strongly to ergosterol concentrations. Perhaps they could refer to a 50% decrease in filipin staining, consistent with the lower ergosterol levels measured spectrophotometrically.

10) Fig 4. Many of the abbreviations in the graphs are not defined in the legend.

11) Line 400. Filipin is a surrogate for PM cholesterol. Therefore, I would change “indicates” to “suggests”.

12) Line 398. What is the significance of observing desmosomes in these mammalian cells?

Reviewer #2: The authors investigate whether nickel overload can influence Cryptococcus neoformans growth and identify a link with sterol regulation, implicated Sre1 and Erg25 as mediating nickel stress resistance. While there are some interesting links between nickel and hypoxia sensitivity identified, and some possible mechanisms proposed through activity of Erg25, these are not fully explored or directly demonstrated. For example, in many places, there is an over-reliance on shared phenotypic outcomes. The work overall is hypothesis generating, rather than a conclusive dissection of function.

While the explanation that Erg25 over-expression can mediate nickel resistance is an interesting insight, and the attempts to predict how changes in Erg25 metal binding would impact function have potential, this reviewer remains unconvinced that nickel directly impacts Erg25 activity in WT or sre1� cells. The alternate interpretation that the observed rescue in Erg25OE cells is mediated by a parallel suppressor activity, for example acting as a nickel sink independent of its native function, does not appear to have been considered. Given that Erg25 acts in a complex with two other EBPs, it remains unclear how Erg25 alone could be influencing nickel and ergosterol-related outcomes.

The sre1� mutant should be reconstituted with the SRE1 gene and key phenotypic analyses performed to confirm that mutant phenotypes are due to loss of SRE1 gene function.

The nic1� mutant should be directly complemented to ensure that phenotypes are directly related to loss of this gene. In addition, conclusions about nic1 having increased nickel relative to WT are not logical (line 151/152). A putative lower affinity nickel transporter, as proposed by Singh et al. (2013), would not explain increased nickel relative to WT, which presumably has both transporters intact.

The authors imply that the sre1� phenotype can be explained by reduced EBP expression, specifically ERG25, and present data that overexpression of ERG25 increases lipid content. However, this hypothesis should be directly tested by measuring expression of these genes in the sre1� mutant (not just by implication of the WT expression changes in nickel and hypoxia) and chemically, for example by adding back the Erg25 intermediate to test for rescue of the phenotype.

Given the hypothesis that nickel and hypoxia exert similar impacts (line 218), comprehensive analysis of the overlap between nickel exposure and hypoxia RNAseq datasets is warranted, beyond the ergosterol pathway.

The introduction overall is discursive and should be substantially revised to focus on the main topic of study. In particular, paragraph three (on bacterial nickel toxicity, and providing conclusions about the work to be performed etc) should be reconsidered. Contractions should not be used in formal writing. Genus names must be specified the first time they are used. I suggest the authors have a senior colleague review the document throughout for overall logical structure and similar errors in scientific writing.

What evidence is there that only one enzyme in the cryptococcus genome that binds to nickel? A reference is needed at the very least. A better approach would be to look for evidence of proteins in the Cryptococcus proteome with the capacity to bind nickel in an unbiased way. The physiological relevance of the chosen nickel and other metal concentrations should be justified.

The author summary overstates the link to mammalian data - there is no evidence that mammalian cells use Erg25 to respond to Nickle stress.

Figure 1B The SRE element should be indicated in the Sre1/DNA interaction.

Figure 1C Micronutrients in their salt form should be described in the figure legend.

Figure S3 What is Co?

Legends are insufficiently detailed. Unnecessary abbreviations should be avoided in figure legends in general.

The methods section "metal assays" is insufficiently detailed. What is the formulation of RPMI plates (supplier information, formulation, etc)? What concentration of the indicated metals were added and in what formulation? How was appropriate pH confirmed?

Appropriate statistical methods must be used throughout. Student's t-test is not appropriate for comparing more than 2 groups. It is unclear what comparisons the p-values are reported for. The p-value must be corrected for multiple comparisons.

**Have all data underlying the figures and results presented in the manuscript been provided?**

Reviewer #1: Yes

Reviewer #2: Yes

PLOS authors have the option to publish the peer review history of their article (what does this mean?). If published, this will include your full peer review and any attached files.

Reviewer #1: No

Reviewer #2: No

---

## [Decision Letter · Decision Letter 1]

30 Aug 2024

Dear Dr Lin,

We are pleased to inform you that your manuscript entitled "Nickel tolerance is channeled through C-4 methyl sterol oxidase Erg25 in the sterol biosynthesis pathway" has been editorially accepted for publication in PLOS Genetics. Congratulations!

Yours sincerely,

Geraldine Butler

Section Editor

PLOS Genetics

Comments from the reviewers (if applicable):

Reviewer's Responses to Questions

**Comments to the Authors:**

Reviewer #1: The authors present a significantly and substantively revised manuscript exploring Ni homeostasis in C. neoformans. They offer new experiments and important text revisions/corrections that make the manuscript much stronger.

**Have all data underlying the figures and results presented in the manuscript been provided?**

Reviewer #1: Yes

PLOS authors have the option to publish the peer review history of their article (what does this mean?). If published, this will include your full peer review and any attached files.

Reviewer #1: No

**Data Deposition**

http://datadryad.org/submit?journalID=pgenetics&manu=PGENETICS-D-24-00338R1

**Press Queries**

---

## [Editor Report · Acceptance letter]

10 Sep 2024

PGENETICS-D-24-00338R1 

Nickel tolerance is channeled through C-4 methyl sterol oxidase Erg25 in the sterol biosynthesis pathway 

Dear Dr Lin, 

We are pleased to inform you that your manuscript entitled "Nickel tolerance is channeled through C-4 methyl sterol oxidase Erg25 in the sterol biosynthesis pathway" has been formally accepted for publication in PLOS Genetics! Your manuscript is now with our production department and you will be notified of the publication date in due course.

With kind regards,

Anita Estes

PLOS Genetics

On behalf of:
